# First-Principles Study on Mechanical and Optical Behavior of Plutonium Oxide under Typical Structural Phases and Vacancy Defects

**DOI:** 10.3390/ma15217785

**Published:** 2022-11-04

**Authors:** Jin-Xing Cheng, Fei Yang, Qing-Bo Wang, Yuan-Yuan He, Yi-Nuo Liu, Zi-Yu Hu, Wei-Wei Wen, You-Peng Wu, Cheng-Yin Zheng, Ai Yu, Xin Lu, Yue Zhang

**Affiliations:** 1Beijing Institute of High Technology, Beijing 100094, China; 2College of Mathematics and Physics, Beijing University of Chemical Technology, Beijing 100029, China

**Keywords:** plutonium oxides, density functional theory, numerical simulation, optical properties

## Abstract

The chemical corrosion aging of plutonium is a very important topic. It is easy to be corroded and produces oxidation products of various valence states because of its 5f electron orbit between local and non-local. On the one hand, the phase diagram of plutonium and oxygen is complex, so there is still not enough research on typical structural phases. On the other hand, most of the studies on plutonium oxide focus on PuO_2_ and Pu_2_O_3_ with stoichiometric ratio, while the understanding of non-stoichiometric ratio, especially for Pu_2_O_3-x_, is not deep enough. Based on this, using the DFT + U theoretical scheme of density functional theory, we have systematically studied the structural stability, lattice parameters, electronic structure, mechanical and optical properties of six typical high temperature phases of β-Pu_2_O_3_, α-Pu_2_O_3,_
γ-Pu_2_O_3,_ PuO, α-PuO_2,_
γ-PuO_2_. Further, the mechanical properties and optical behavior of Pu_2_O_3-x_ under different oxygen vacancy concentrations are analyzed and discussed in detail. The result shows that the elasticity modulus of single crystal in mechanical properties is directly related to the oxygen/plutonium ratio and crystal system. As the number of oxygen vacancies increases, the mechanical constants continue to increase. In terms of optical properties, PuO has the best optical properties, and the light absorption rate decreases with the increase of oxygen vacancy concentration.

## 1. Introduction

The study of plutonium and its oxides has received continuous attention due to the complexity of the 5f state localized local/discrete domain of plutonium (Pu), resulting in a complex and rich phase diagram of plutonium oxide [1,2,3]. Plutonium and its oxide are important components of mixed oxide fuel in fast breeder reactors and advanced heavy water reactors [4,5,6,7,8,9,10,11], which plays an important role in the nuclear industry. Of course, they may also be stored as radioactive waste for long periods [12]. The phase diagram of plutonium oxide is intricate by the presence of multiple suboxides. It is recognized that the initial oxidation of plutonium produces trivalent sesquioxides (Pu_2_O_3_), followed by the trivalent dioxides (PuO_2_). The coexistence of PuO_2_ and plutonium leads to a reduction reaction of 3PuO_2_ + Pu → 2Pu_2_O_3_, which is feasible in thermodynamics and results in a Pu_2_O_3_/Pu interface. In plutonium oxide layer, other non-stoichiometric plutonium oxides PuO_2-x_ (0 < x < 0.5) can also be formed. The ratio of Pu_2_O_3_ to PuO_2_ is very sensitive to temperature and other external environments [1,13,14,15,16,17]. Generally, the base state of plutonium suboxide is the hexagonal structure β-Pu_2_O_3_ (P3m1), in addition, there is another oxide with a body-centerd cubic structure (α-PuO_1.52_) [18]. Further studies revealed that the cubic phase sesquioxide-plutonium (α-Pu_2_O_3_) also exists stably.α-Pu_2_O_3_ consists of 32 plutonium atoms and 48 oxygen atoms (Ia-3 space group) but it has not been prepared as a single-phase compound. This is because α-Pu_2_O could only be stable below 300 °C and can be partially reduced on PuO_2_ by the mixture of PuO_1.52_ and PuO_1.98_ [19] at high temperature, and then cooled to room temperature. Above 650 °C, 18 low-valence oxides between PuO_1.61_ and PuO_2_ (i.e., PuO_1.61+y_) mainly exist in a single-phase structure and can be prepared by reducing PuO_2_ to carbon or hydrogen in vacuum [19]. So far, their final structure of this phase has yet to be determined. However, above 650 °C, 18 low-valence oxides between PuO_1.61_ and PuO_2_ (i.e., PuO_1.61+y_) are likely to be the subchemometric plutonium dioxide (PuO_2_) phase. Additionally, the structure of PuO_1.61_ is closely related to cubic PuO_1.52_. Therefore, it can be preliminarily described as the same cubic structure with 32 metal atoms and 51/52 oxygen atoms in a single cell. In contrast, in XRD measurements above 300 °C, [19] only strong pseudo-face-centered cubic reflections can be observed. Above 650 °C, PuO_2_ can survive in a super-poor oxygen state, where the oxygen stoichiometry ratio can be reduced to PuO_1.7_ or even PuO_1.6_ [20]. The presence of oxygen also confirmed from density measurements at 750 °C vacancies, rather than the formation of interstitial plutonium ions.

Theoretically, it is a challenge to study plutonium oxides within the framework of electronic structure calculation based on standard Density Functional Theory (DFT). Due to the narrow *f*-band, close proximity to *d* and *s* orbitals in the metallic element plutonium, the phenomena of significant dynamic charge rise and fall exhibit in metallic bonds [21,22,23,24]. In the standard framework of local density generalization and generalized gradient generalization, the local effect of f- electrons caused by strong electron-electron interaction is not captured. To overcome this difficulty, different methods such as calculations involving self-interaction corrections (SIC) [25,26], mixed exchange correlation functions [27,28] or intra-atomic Coulomb interactions (Hubbard’s U-parameters) [29,30,31,32,33] have been proposed and applied to study the two stoichiometric oxides (PuO_2_, Pu_2_O_3_) in terms of structural, electronic, thermodynamic and optical properties [34,35,36]. Raymond et al. [37] calculated the electronic properties of the native (11$\bar{2}$0) surfaces of β-Pu_2_O_3_ and found that the antiferromagnetic structure was more stable than the ferromagnetic structure. Lu et al. [38] used DFT + U method to determine the stability of charged defects in PuO_2_. Petit et al. [25] analyzed the electronic structures of PuO_2.25_, PuO_1.75_ and PuO_2_ using the SIC approximation. Agarwal et al. [39] used the Gibbs Free Energy Model to calculate the phase diagram of PuO_2-x_ (0 < x < 0.5) and observed the mixed-phase gap behavior of PuO_2_ and PuO_1.7_. The results showed that the pure up Pu-O system can reduce to 60% of the plutonium to the +3 oxidations state while maintaining the face-centerd cubic (fcc) structure. However, there are still few studies on high-temperature phase stability, electronic structure and formation mechanism of non-chemiscale PuO_2_ and Pu_2_O_3_, especially for typical high-temperature phases and Pu_2_O_3-x_ (0 < x < 1) with different concentrations of oxygen vacancy defects. In this paper, the mechanical and optical properties of six typical high-temperature phase structures of Pu oxides are systematically discussed by DFT + U method, including β-Pu_2_O_3_, α-Pu_2_O_3_, γ-Pu_2_O_3_, PuO, α-PuO_2_, γ-PuO_2_. Considering the obvious influence of vacancy defects on the properties of the system [40], we have also focused on the mechanical and optical properties of β-Pu_2_O_3_, structures with different concentrations of oxygen vacancy defects.

## 2. Computational Methods

First-principles calculation is carried out using the VASP (Vienna ab initio simulation package) software package [41]. The generalized-gradient approximation (GGA) with Perdew-Burke-Emzerhof (PBE) form [42] was used to describe the exchange-correlation functional. Setting the cut-off energy as 540 eV, we take the 6s^2^7s^2^6p^6^6d^2^5f^4^ of plutonium and the 2s^2^2p^4^ electrons of O as valence electrons [43] to participate in the calculation. The in situ repulsive energy of the plutonium 5f orbital electrons is considered in the calculations, and the values of U and J are, respectively, 4. 75 eV and 0. 75 eV [44]. For the calculations of the β-Pu_2_O_3_, α-Pu_2_O_3_, γ-Pu_2_O_3_, PuO, α-PuO_2_, and γ-PuO_2_ and the Monkhorst-Pack k-point grid for the Brillouin zone integral were sampled on 10 × 10 × 6(β-Pu_2_O_3_), 7 × 7 × 7(α-Pu_2_O_3_), 10 × 10 × 7(-Pu_2_O_3_), 8 × 8 × 8(PuO), 8 × 8 × 8(α-PuO_2_), 7 × 7 × 10(γ-PuO_2_) [7]. All structures were completely relaxed and the residual force was less than 0.01 eV/Å.

Spin-polarized are included in structural optimizations. Here, we consider three possibilities for the magnetic states: nonferromagnetic (NM), ferromagnetic (FM) and antiferromagnetic (AFM) for all the oxides, see Appendix A. In the following FM calculations, we use the collinear 1-k structure where the atomic spin moment is along the [001] direction. Energy formation is the total energy of **a** crystal minus the energy of the individual atoms contained in the crystal. In order to analyze the structural stability of the six plutonium oxides, their energy formation (*E_form_*) has been calculated,
(1)$$E_{form}= [E_{tot}\;\left(Pu_{m}O_{n}\right)-mE_{tot}\left(Pu\right)-nE_{tot}\;\left(O\right)]⁄\left(m+n\right))$$
where m and n are the number of *Pu* and *O* atoms in the supercell. Respectively, $E_{tot}\;\left(Pu_{m}O_{n}\right)$, $E_{tot}\left(Pu\right)$ and $E_{tot}\;\left(O\right)$ are the total energy of per atom in their solid state. In general, the lower the formation energy, the more stable the solid solution. Additionally, the lower the formation energy of impurities, the more efficient the doping process.

For the six plutonium oxides, three independent elastic stiffness constants C_11_, C_12_, C_44_ were calculated by Voigt Method and obtained the Bulk Modulus (B), Shear Modulus (G), Young’s Modulus (E) and Poisson’s Ratio (υ). It is calculated as follows:(2)$$B=\frac{1}{3}\; [C_{11}+2C_{12}\;]\;$$
(3)$$G=\frac{1}{5} [C_{11}-C_{12}+3C_{44}]\;$$
(4)$$E=\frac{9BG}{3B+G}$$
(5)$$\upsilon =\frac{3B-2G}{2\left(3B+G\right)}$$

Other than that, Debye temperature is also closely related to the elastic constant. Based on the estimation of the average speed of sound ($v_{m}$) in a given material, the commonly used method of calculating Debye temperature is as follows:(6)$$\theta_{D}=\frac{h}{k}{ [\frac{3n}{4\pi }{\left(\frac{N_{A}\rho }{M}\right)}^{1}]}^{\frac{1}{3}}v_{m}$$

As shown in the formula above, h, k, $N_{A}$, n, M and ρ represent Planck’s Constant, Boltzmann’s Constant, Avogadro’s Number, the atoms, molecular weight and density of organic cations. The average sound velocity ($v_{m}$) can be calculated by shear velocity ($v_{t}$) and compression velocity ($v_{l}$), respectively. $v_{m}$ is related to $v_{t}$ and $v_{l}$ and are listed as follows:(7)$$v_{t}=\sqrt{G/\rho }$$
(8)$$v_{l}=\sqrt{\left(B+\frac{3}{4}G\right)/\rho }$$
(9)$$v_{m}={ [\frac{1}{3}\left(\frac{2}{v_{1}{}^{3}}+\frac{1}{v_{t}{}^{3}}\right)]}^{-\frac{1}{3}}$$

The optical properties of a compound are described by a complex function. The imaginary part of the dielectric tensor is determined by the sum of the following empty band states:(10)$$\varepsilon_{2}\left(\omega \right)=\frac{2\pi e^{2}}{V\varepsilon_{0}}\underset{k,v,c}{\sum }\delta \left(E_{k}^{c}-E_{k}^{v}-\hbar \omega \right){\left|\langle \Psi_{k}^{c}|\mu \cdot r{|\Psi }_{k}^{v}\rangle \right|}^{2}$$

$\varepsilon_{0}$ is the vacuum dielectric constant, V is the crystal volume. v and c, respectively, denotes the valence and conduction bands. ℏω is the energy of the incident phonon. u is defined as the vector of incident electric field polarization. μ·r is the momentum operator. $\Psi_{k}^{c}$ and $\Psi_{k}^{v}$ are the wave functions of point k the conduction and valence bands, respectively. The real part of the dielectric tensor is proposed by the well-known Kramers-Kronig relation:(11)$$\varepsilon_{1}\left(\omega \right)=1+\frac{2}{\pi }P\underset{k,v,c}{\sum }\int_{0}^{\infty }\frac{\varepsilon_{2}\left(\omega^{\prime }\right)\omega^{\prime }}{{\omega^{\prime }}^{2}-\omega^{2}+i\eta }d\omega^{\prime }$$

## 3. Results and Discussion

### 3.1. Crystal Structure Analysis

Plutonium oxide has multiple structures, and Figure 1 shows the crystal structure parameters and local charge density of six plutonium oxides. Pu_2_O_3_ has three forms, with β-Pu_2_O_3_ containing in the tripartite crystal system (P3m1 space group), α-Pu_2_O_3_ containing in the cubic crystal system (Pn3m space group) and γ-Pu_2_O_3_ containing in the tetragonal crystal system (P4m2 space group). PuO belongs to the cubic crystal system (Fm3m space group). The PuO_2_ exists in two forms and α-PuO_2_ belongs to the cubic crystal system (Fm3m space group) and γ-PuO2 belongs to the tetragonal crystal system (P42/mnm) space group. Through structural optimization, their equilibrium lattice constants were calculated and listed in Table 1.

The formation energies of the six oxides were calculated according to Equation (1). As shown in Table 1, the formation energies of all six plutonium oxides are negative, which indicates that the formation of the compounds is exothermic. β-Pu_2_O_3_ and γ-Pu_2_O_3_ have the same lattice constants a, b, c and bond angles α, β, γ, but β-Pu_2_O_3_ has a lower formation energy. β-Pu_2_O_3_ structure is more energetically stable than γ-Pu_2_O_3_. By comparison, the formation energies of the two PuO_2_ structures show that the α-PuO_2_ structure is more stable than the γ-PuO_2_ structure.

### 3.2. Band Structure and Density of Electronic States

As shown in Figure 2, we have calculated the energy band diagrams and total electron density of states (TDOS) diagrams to investigate the properties of the six plutonium oxides. Based on the properties of plutonium, spin orbit coupling is considered in our calculations. From Figure 2a–f, we can see that electrons with the same spin state are concentrated near the Fermi energy level, and at energies below −2 eV. The TDOS is contributed by both spin-up and spin-down electrons, and at energies −2 eV < E < 2 eV, β-Pu_2_O_3_ and *α*-PuO_2_ is only contributed by spin up states in −2 eV to 2 eV, but with little effect of the spin-down electronic states for β-Pu_2_O_3_ and *α*-PuO_2_ in −2 eV to 2 eV. Additionally, at energies greater than 2 eV, the TDOS is mainly contributed by spin-down electrons. In view of this, we can filter the electrons with different spin states according to different energy ranges, which is of great importance for quantum applications.

In general, the stability of the material structure was determined by the position of the value of the electronic density of states (DOS) at the Fermi energy level. For the metallic system, the distribution of electronic DOS will span the Fermi level and have a great impact on the properties of materials. In Figure 2, the red line indicates spin up, and the blue line indicates spin down. We can see that the spin up electron density of the six plutonium oxides crosses the Fermi level. However, only the electron densities of Figure 2b–d cross the Fermi level for spin down electrons, while for spin down electrons, the band gaps of Figure 2a,e,f are 4.46 eV, 4.85 eV and 4.11 eV. We have calculated the electronic projected density of states (PDOS) of plutonium oxides, as shown in Figure 3. It shows that the overall picture is similar and the electron contribution to the density of states can be discussed in three intervals from Figure 3a–f. At energies below −1 eV, for β-Pu_2_O_3_, α-Pu_2_O_3_, α-PuO_2_ and γ-PuO_2_, the DOS is mainly contributed by the p-orbitals of the oxygen atom, while the orbital contributions of γ-Pu_2_O_3_ and PuO are similar. Near the Fermi energy level, it can be clearly seen that the f-orbiting electrons of the plutonium atom play a major role and are contributed by the separate spin-up electron. We believe that for the six plutonium oxides, the f-orbital electrons of Pu atom mainly contribute to the DOS at the Fermi level. The contribution of O atom is small, because the energy of 2p electron in the outermost layer of oxygen atom is smaller than 5f of Pu. At the same time, the 2p orbital electron of oxygen atom has a large expansibility which means the degree of electron dispersion is strong. From each figure, we can see an obvious spike in electronic density of states, which is mainly caused by the strong localization of 5f orbital electrons of Pu atom.

### 3.3. Mechanical and Optical Properties

Table 2 shows the mechanical constants C_ij_ of plutonium oxides. For a mechanically stable material with a cubic structure [45,46], the independent elastic constants need to meet the following three criteria C_11_ > |C_12_|, C_44_ > 0 and C_11_+2C_12_ > 0. The Bulk modulus (B), Young’s modulus (E), Shear modulus (G) and Poisson’s rati (υ) of several plutonium oxides are given in Table 3. Brittleness and ductility can be determined by the value of B/G. When B/G is small, the material is brittle; otherwise, it is ductile. From the calculations, it is possible to see that the B/G value of β-Pu_2_O_3_ is large and ductile. The bulk modulus (B) indicates incompressibility, and Table 3 shows that PuO has the greatest incompressibility followed by α-PuO_2_. The modulus of elasticity E is an important parameter to characterize material stiffness, which mainly reflects compressive strength of the material. From Table 3, we find that α-PuO_2_ has the highest stiffness. The Shear modulus (G) is a measure of the ability of a material to resist shear deformation, and Table 3 shows that α-PuO_2_ has the largest Shear modulus, so it has the strongest shear deformation resistance. In addition, the smaller Poisson’s ratio is, the more brittle the material is. The result is consistent with our analysis above.

The study of optical properties of materials is one of the most effective techniques for analyzing various physical properties related to the electronic structure of materials. The dielectric function, optical absorption spectrum and reflection spectrum of plutonium oxide have been calculated and analyzed. The imaginary and real parts of the dielectric functions of plutonium oxide with different structures are shown in Figure 4. In the low energy region, the real part of the dielectric function decreases to low value with the energy increasing and then increases slowly and then remains essentially constant. The imaginary part of the dielectric function decreases as the energy increases in low energy region, a new peak at energies above 20 eV, but the peak decreases. The curve for PuO is the most unusual, with a unique peak in the low energy band. The absorption and reflection spectra of different structures of plutonium oxide are shown in Figure 4c,d. It shows that the photon absorption of plutonium oxides diminishes with the increasing wavelength. In the visible range, the absorption decreases and then increases steadily with the increasing wavelength. In the UV region, it shows that the different plutonium oxides have different peaks of wavelength absorption. Compared with other plutonium oxides, the absorption of some photons in the violet and UV regions is the strongest for PuO and forms the two highest peaks evident, with the plutonium oxides reflecting more significantly in the visible region and stable in the infrared region as the wavelength increases. PuO remains the most unusual, with two distinct peaks. In Figure 4, the un-reported metastable oxide of α-Pu_2_O_3_ presents the highest dielectric properties. Light absorption spectra or reflection spectra also present different absorbed properties.

### 3.4. Mechanical and Optical Properties at Different Oxygen Vacancy Concentrations

After analyzing the previous research, we found that β-Pu_2_O_3_ is the most stable among the six plutonium oxides structures, and at the same time its moldability and forgeability are better. This is more meaningful and valuable for the research. The current research on plutonium oxides mainly focuses on PuO and PuO_2_, but there are few studies on Pu_2_O_3_ under high temperature phase. So, we expanded the cell of β-Pu_2_O_3_ by 2 × 2 × 2 times along the crystal direction to obtain the structure of β-Pu_16_O_24_. We studied the properties under different oxygen vacancy concentrations after rounding off its oxygen atoms in the ratio of 1:2:4:6 and obtained the electronic PDOS maps for (a) β-Pu_16_O_23_, (b) β-Pu_16_O_22_, (c) β-Pu_16_O_20_ and (d) β-Pu_16_O_18_. The above is shown in Figure 5. In the energy region below −4 eV, it is mainly the *p* orbitals of O atoms that contribute to the major DOS, followed by the *d* and *f* orbitals and *p* orbitals of Pu atoms. There are no electronic states that are occupied between energies −4 eV and −1 eV. As the energy continues to 0 eV, DOS is mainly influenced by f-orbitals of Pu atom, followed by d orbitals of Pu atom, and there is almost no contribution from the oxygen atom. Secondly, the contribution of the *f* orbitals of Pu to the DOS decreases as the oxygen vacancy concentration increases.

Debye temperature is an important physical quantity that reflects the interatomic bonding forces. The melting point of a material is positively correlated with the interatomic bonding force. The Debye temperature increases with the values of Pu/O increasing; when the value of Pu/O varies from 0.696 to 0.727, and decreases with the values of Pu/O increasing. When it varies from 0.727 to 0.889, β-Pu_16_O_22_ has the strongest interatomic bonding and the highest Debye temperature among the four calculated oxygen vacancy plutonium oxide structures. The crystal structure is more complex, the anharmonic degree of lattice vibrations is greater. If the lattice wave scattering is larger, the mean free path of phonons is smaller, and the thermal conductivity is lower. Results show that the lattice thermal conductivity of β-Pu_16_O_23_ is the lowest among the four oxygen vacancy plutonium oxides, as shown in Table 4.

To further investigate the properties of β-Pu_2_O_3_ at different oxygen vacancy concentrations, we calculated the transverse sound velocity (ν_t_), longitudinal sound velocity (ν_l_) and average sound velocity (ν_m_) of Pu_16_O_23_, Pu_16_O_22_, Pu_16_O_20_ and Pu_16_O_18._ The result is shown in Figure 6a. The calculated Bulk modulus (B), Shear modulus (G), Young’s modulus (E) and Poisson’s ratio (υ) were compared with the elastic constants of Pu_32_O_62_, Pu_32_O_60_, Pu_32_O_56_ and Pu_32_O_52_ calculated by Ghosh et al. [47], which is shown in Figure 6b. We found that Pu_16_O_20_ is more special and manifested mechanical constants of C_11_ > C_33_, C_66_ > C_44_, while the values of C_11_, C_33_, C_66_, C_44_ kept increasing as the number of oxygen vacancies of the other three oxides increases, which it always shows C_11_ < C_33_, C_66_ < C_44_. Comparing with the values of C_ij_ calculated by Ghosh et al. [47], who studied four plutonium oxides and found that the values of C_11_, C_33_, C_44_, and C_66_ decrease gradually as oxygen vacancies increase. From Figure 6a, it shows that the transverse sound velocity (ν_t_), longitudinal sound velocity (ν_l_) and mean sound velocity (ν_m_) increase with the values of Pu/O increasing when the value of Pu/O varies from 0.696 to 0.727. They decrease with the values of Pu/O increasing when the value of Pu/O varies from 0.727 to 0.889 f. Shear modulus (G) is the ratio of shear stress to shear strain, which represents the ability of materials to resist shear strain. If the modulus is larger, the material is harder. The ratio of G to B (G/B) indicates the brittleness and toughness of material, and it can be found that Pu_16_O_20_ has the best brittleness and toughness of the four plutonium oxides we studied, while Pu_16_O_23_ has the smallest G/B value. That means the brittleness and toughness of the material gradually increases as the value of Pu/O increases from 0.696 to 0.8, and is predicted to decrease when the values are greater than 0.8. It can be observed that Poisson’s ratio (υ) gradually decreases as the value of Pu/O gradually increases from 0.69 to 0.8 and then gradually increases again when the values are greater than 0.8. Compared with the case calculated by Ghosh et al. [47], the study reveals that the oxygen vacancy profile of β-Pu_16_O_24_ does not vary in a single way.

The imaginary and real parts of dielectric function of plutonium oxide at different oxygen vacancy concentrations are shown in Figure 7a,b. In the low energy region, the values of both real and imaginary parts of the dielectric function decrease. As oxygen atoms decrease, the energy gradually increases. The values of the real and imaginary parts of the dielectric function increase as oxygen atoms decrease. Figure 7c shows the optical absorption spectrum of plutonium oxide at different oxygen vacancy concentrations. The main absorption peak is around 40 nm (in the vacuum UV region), which indicates that most electrons can transition from the valence band to the conduction band by absorbing a small amount of energy corresponding to the system band gap. When the wavelengths are less than 700 nm, the absorption of light of the plutonium oxide increases with the decreasing numbers of oxygen atoms at the same wavelength of incident light. When a turnaround is near 700 nm, the absorption of the light of the plutonium oxide decreases with the decrease of oxygen atoms. Figure 7d shows the reflection spectrum of plutonium oxide at different oxygen vacancies. The reflection values are larger in the visible region from 380–790 nm. It shows that when the wavelengths are less than 700 nm, at the same wavelength, the light absorption of plutonium oxide increases with the decrease of oxygen atoms. However, there is a turnaround and the light absorption of plutonium oxide decreases as oxygen atoms decrease when wavelengths are around 700 nm. The change in the negative dielectric function is well supported the conclusions obtained in optical absorption spectrum and absorption spectrum.

## 4. Conclusions

After systematically investigating the electronic structure, mechanical and optical properties of plutonium oxides β-Pu_2_O_3_, α-Pu_2_O_3_, γ-Pu_2_O_3_, PuO, α-PuO_2_, γ-PuO_2_ and β-Pu_2_O_3-x_ at different oxygen vacancy concentrations by using the first principles, the results show that the vicinity of the Fermi energy level is mainly determined by the spin-up electron state of 5f orbital of Pu atom, while the spin-down electronic states of the 5f orbitals are at higher energy levels. This is useful for distinguishing and screening the electrons of different spin states. The higher the oxygen/plutonium ratio, the greater the bonding between the crystalline atoms. The elastic modulus is higher, the material is harder. When the oxygen/plutonium ratio is 1.5, the elastic modulus of the γ crystal system is higher than β crystal system. The α crystal system is the smallest. However, at an oxygen/plutonium ratio of 2, the elastic modulus of α crystal system is higher than γ crystal system. For the study of the UV light of any of the plutonium oxides, PuO has the strongest absorption in the UV region and the most unusual reflective properties, which there are two reflective peaks in the entire waveband.

## Figures and Tables

**Figure 1 materials-15-07785-f001:**
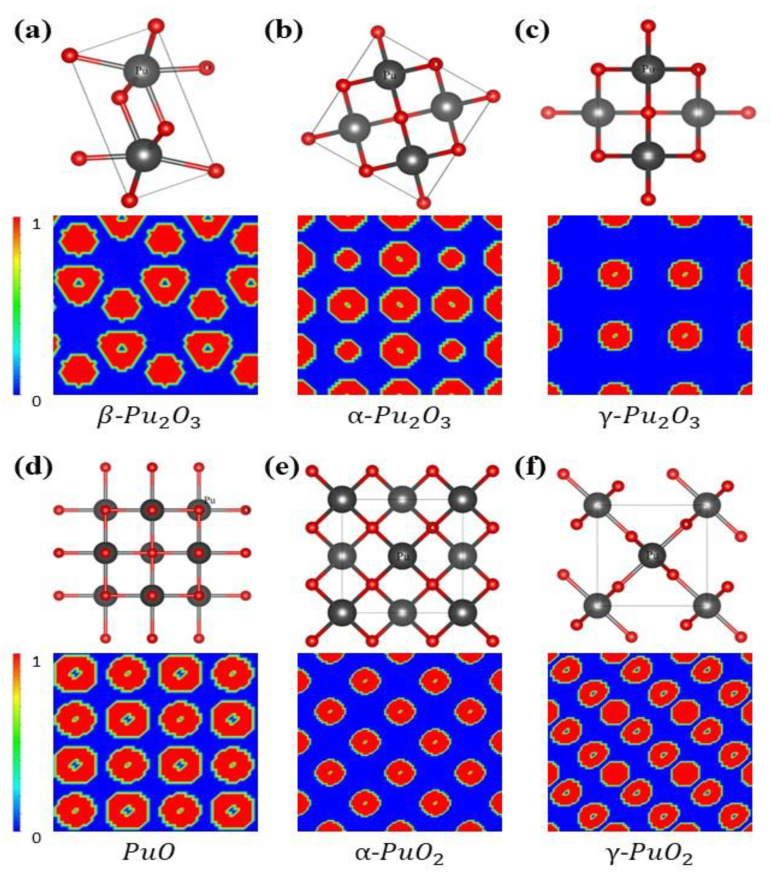
Optimized geometries of (**a**) β-Pu_2_O_3_, (**b**) α-Pu_2_O_3_, (**c**) γ-Pu_2_O_3_, (**d**) PuO, (**e**) α-PuO_2_, (**f**) γ-PuO_2_ are shown in the figure above. The gray atoms indicate the Pu atoms. The red atoms indicate O atoms. Their local electronic density states are also listed. Pu and O atoms are represented in blue and red color, respectively.

**Figure 2 materials-15-07785-f002:**
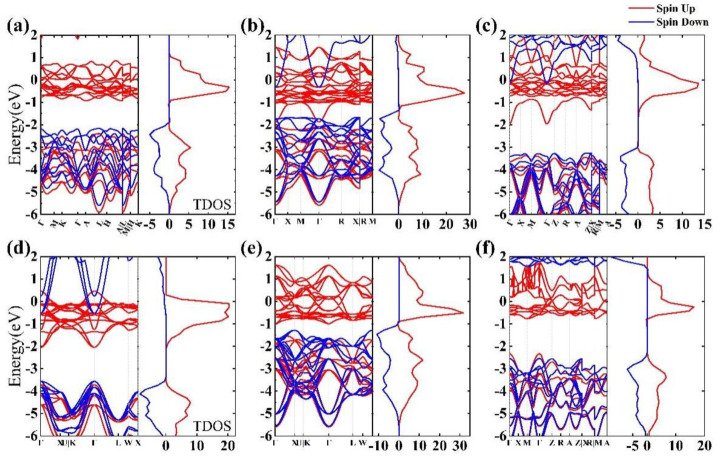
The upper left figure shows the electronic structure band, and the upper right figure shows the total electron density of states (TDOS) of (**a**) β-Pu_2_O_3_, (**b**) α-Pu_2_O_3_, (**c**) γ-Pu_2_O_3_, (**d**) PuO, (**e**) α-PuO_2_, (**f**) γ-PuO_2_. The red line is represented by electrons that spin downwards and the blue line is represented by electrons that spin upwards. The location with an energy of 0 represents the Fermi level.

**Figure 3 materials-15-07785-f003:**
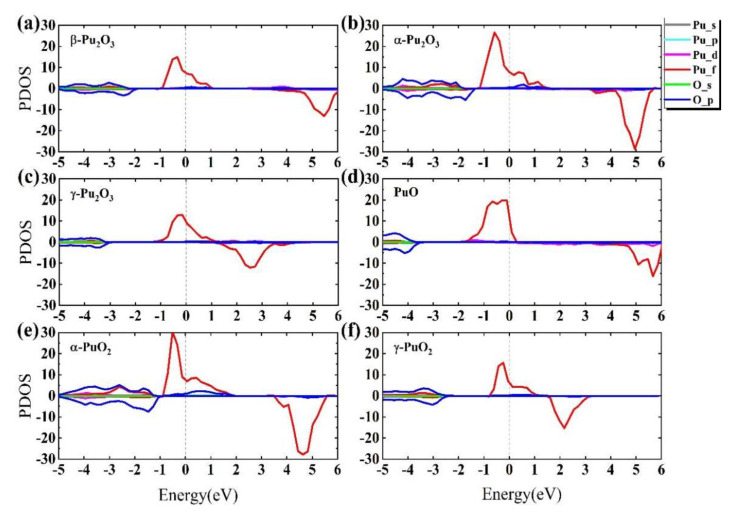
The electronic projected density of states (PDOS) of (**a**) β-Pu_2_O_3_, (**b**) α-Pu_2_O_3_, (**c**) γ-Pu_2_O_3_, (**d**) PuO, (**e**) α-PuO_2_, (**f**) γ-PuO_2_.

**Figure 4 materials-15-07785-f004:**
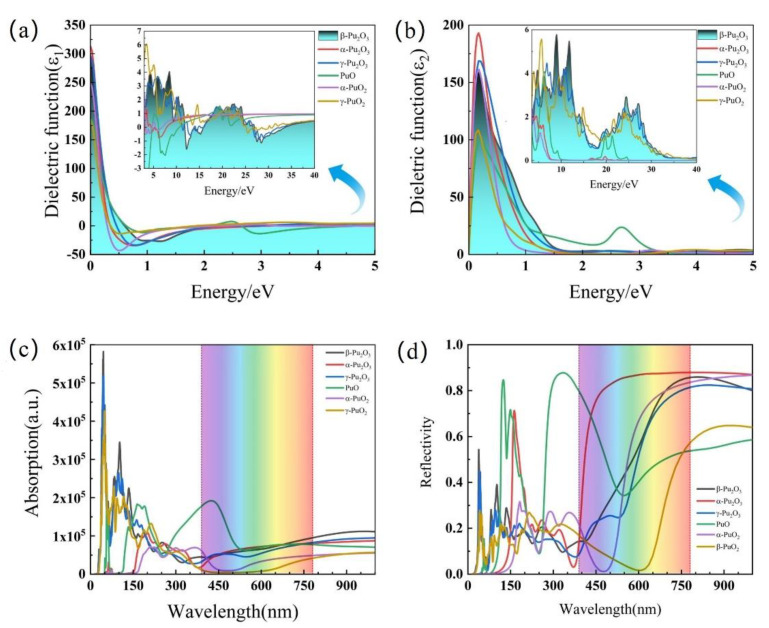
Real $\varepsilon_{1}\left(\omega \right)$, imaginary $\varepsilon_{2}\left(\omega \right)$ components of the dielectric function in (**a**,**b**). (**c**) shows light absorption spectra and (**d**) shows reflection spectra of studied oxide.

**Figure 5 materials-15-07785-f005:**
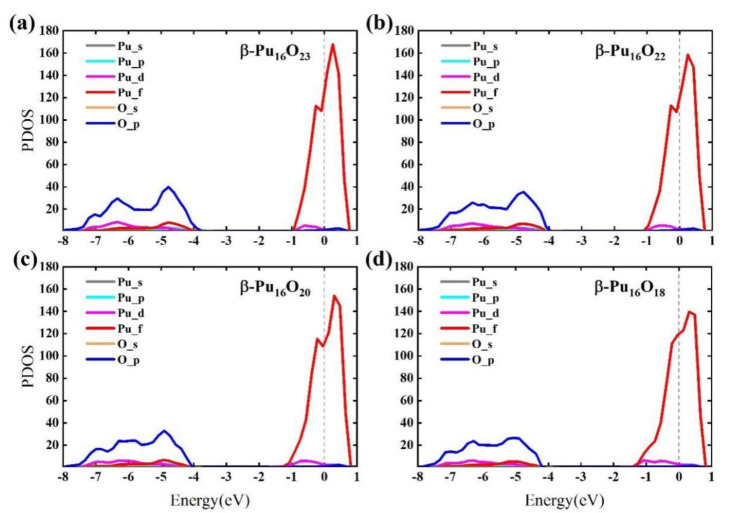
The electronic Pre-Departure Orientation Seminar (PDOS) of (**a**) β-Pu_16_O_23_, (**b**) β-Pu_16_O_22_, (**c**) β-Pu_16_O_20_, (**d**) β-Pu_16_O_18_.

**Figure 6 materials-15-07785-f006:**
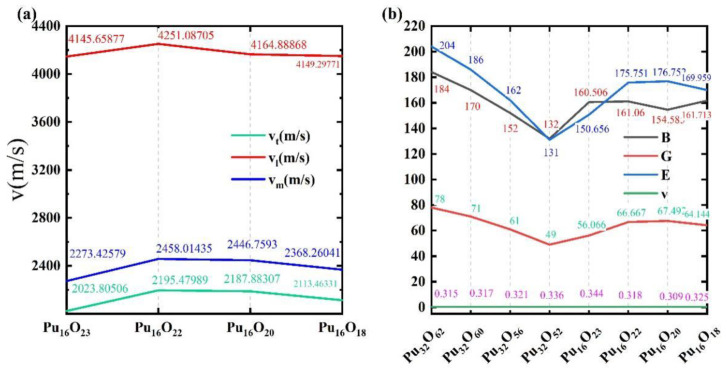
(**a**) Values of average sound velocity (ν_m_), longitudinal wave velocity (ν_l_), shear wave velocity (ν_t_) of Pu_16_O_23_, Pu_16_O_22_, Pu_16_O_20_, Pu_16_O_18_ using VASP. (**b**) Values of the Young’s modulus (E), Bulk modulus (B) and Shear modulus (G) (in GPa), and Poisson’s ratio (υ
) of Pu_16_O_23_, Pu_16_O_22_, Pu_16_O_20_, Pu_16_O_18_ are compared with the values of Pu_32_O_62_, Pu_32_O_60_, Pu_32_O_56_, Pu_32_O_52_ calculated by Ghosh [47,48].

**Figure 7 materials-15-07785-f007:**
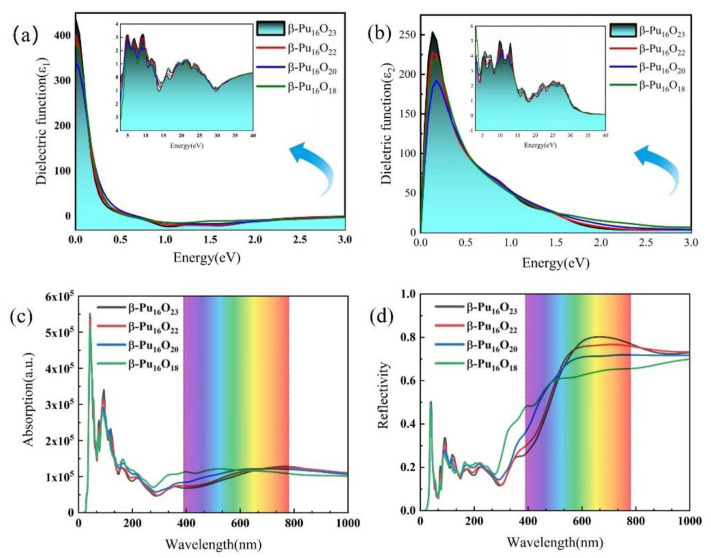
Real $\varepsilon_{1}\left(\omega \right)$, imaginary $\varepsilon_{2}\left(\omega \right)$ components of the dielectric function, light absorption spectra and reflection spectra of β-Pu_16_O_23_, β-Pu_16_O_22_, β-Pu_16_O_20_ and β-Pu_16_O_18._ (**a**) Real part of dielectric function. (**b**) Imaginary part of dielectric function. (**c**) Absorption spectra. (**d**) Reflection spectra.

**Table 1 materials-15-07785-t001:** Lattice parameters of β-Pu_2_O_3_, α-Pu_2_O_3_, γ-Pu_2_O_3_, PuO, α-PuO_2_, γ-PuO_2_, include formation energy, lattice constant and bond angle α, β, γ.

Plutonium Oxide	Formation Energy (eV)	a (Å)	b (Å)	c (Å)	α (°)	β (°)	γ (°)
β ** $-Pu_{2}O_{3}$ **	−3.411	3.76	3.76	5.97	90	90	120
α ** $-Pu_{2}O_{3}$ **	−3.357	5.38	5.38	5.38	90	90	90
γ ** $-Pu_{2}O_{3}$ **	−3.401	3.76	3.76	5.97	90	90	120
** PuO **	−2.808	4.97	4.97	4.97	90	90	90
α ** $-PuO_{2}$ **	−3.673	3.76	3.76	5.97	90	90	120
γ ** $-PuO_{2}$ **	−3.507	3.76	3.76	5.97	90	90	120

**Table 2 materials-15-07785-t002:** Values of mechanical constants C_ij_ of each lead oxide.

	C_11_	C_12_	C_13_	C_14_	C_33_	C_44_	C_66_
β ** $-Pu_{2}O_{3}$ **	253.275	122.725	151.414	8.55	228.163	−8.55	65.275
α ** $-Pu_{2}O_{3}$ **	305.341	4.798	4.798	-	305.341	−17.313	−17.313
γ ** $-Pu_{2}O_{3}$ **	226.968	127.209	92.974	-	259.606	54.821	96.290
** PuO **	425.817	112.785	112.785	-	425.817	45.283	45.283
α ** $-\mathrm{Pu}O_{2}$ **	3925.219	2276.845	2276.845	-	3925.219	65.254	65.254
γ ** $-\mathrm{Pu}O_{2}$ **	4966.565	5302.567	112.485	-	572.217	−433.486	445.562

**Table 3 materials-15-07785-t003:** Values of the Young’s modulus (E), Bulk modulus (B) and Shear modulus (G) (in GPa), and Poisson’s ratio (v) of each plutonium oxide.

Plutonium Oxide	Bulk Modulus (GPa)	Young’s Modulus (GPa)	Shear Modulus (GPa)	Poisson’s Ratio
β-$Pu_{2}O_{3}$	176.2	155.78	57.58	0.35
α-$Pu_{2}O_{3}$	104.98	128.82	49.72	0.3
γ-$Pu_{2}O_{3}$	148.87	171.67	65.63	0.31
** PuO **	217.13	205.43	76.52	0.34
α-**Pu**$O_{2}$	209.49	281.5	110.3	0.28
γ-**Pu**$O_{2}$	175.13	153.61	56.73	0.35

**Table 4 materials-15-07785-t004:** Values of the Debye temperature and lattice thermal conductivity of the studied structures.

Plutonium Oxide	Debye Temperature (K)	Lattice Thermal Conductivity
β-$Pu_{16}O_{23}$	285.864	0.8124
β-$Pu_{16}O_{22}$	307.852	0.8501
β-$Pu_{16}O_{20}$	303.668	0.8250
β-$Pu_{16}O_{18}$	301.879	0.8535

## Data Availability

The data presented in this study are available on request from the corresponding authors.

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
