# Peer review of "First-Principles Study on Mechanical and Optical Behavior of Plutonium Oxide under Typical Structural Phases and Vacancy Defects"

_materials, 2022, doi:10.3390/ma15217785_

Round 1

Reviewer 1 Report

Cheng at all. using the density functional theory to investigate the structural stability, lattice parameters, electronic structure, mechanical and optical properties of six Pu oxide phases. The variations of the mechanical and optical properties in the presence of oxygen vacancies is also considered. A correlation between the oxygen / plutonium ratio and mechanical properties is found. The best optical performance is calculated for PuO. In general, the manuscript is confusing, and it is very difficult to understand which are the relevant information that the authors gain from each section. Some examples:

a   1) In section 3.2. Energy Band Structure and Electronic Density of States, the band structures and DOS/PDOS are reported but which kind of relevant information is obtained from them?

b   2) In the 3.3 Section, the authors wrote “Only the β-Pu2O3 structure meets all three requirements to indicate that only β-Pu2O3 is mechanically stable.” This is the result, and then? What kind of relevant information about material properties can be derived from this sentence, and how can this sentence be related to the formation energies in Table 1? Does the fact that β-Pu2O3 is the only mechanically stable species also imply that it is the only species present under the experimental conditions? In this case, the subsequent calculations on all species are meaningless.

c  3) In section 3.3, the trends in Figure 4 are described, but what the implications are on the properties of the different trends is not reported.

For all these reasons I suggested to not publish the manuscript in the present form. Major revisions are eventually necessary to better highlight the aim and the implications of each calculation on the different systems.

In addition, minor revisions:

1   1) In Table 1, the formation energy unit is missing

2   2) In Figure 2, it would be better to indicate in the caption which is the atom involved in the PDOS. But in the text is indicated that Figure 2 reported total DOS, thus the authors should correct this incongruence.

3  3) Authors wrote that “The total electron density of states is contributed by both spin-up and spin-down electrons, and at energies -2 eV < E < 2 eV, the total electron density of states for β-Pu2O3 and α-PuO2 is essentially contributed by spin-up electrons.” But in this range this sentence in valid for all species.

4  4) Authors wrote that “From Figure 2a, we can find that β-Pu2O3 has the lowest total DOS at the Fermi energy level and indicates that the structure of β-Pu2O3 is more stable compared with several other structures.” But this does not seem to be true. If the 0 in y scale on the left is the Fermi level, the lowest value of the total DOS on the x axes seems to be found for the PuO specie as clearly reported in Figure 3.

5   5) The PDOS acronymous is not “Pre-Departure Orientation Seminar (PDOS)” as reported in the caption of Figure 3. This error is particularly serious in the reviewer's opinion because it conveys the idea that it is not clear to all the authors (or at least to those who wrote this caption) what kind of calculations took place.  

6   6) In the discussion between Figures 2 and 3 authors compared “γ-Pu2O3 and γ-Pu2O3”. Probability there is a typo.

Author Response

*******************************************************************

--------------REPLY TO THE FIRST REVIEWER-------------

Reviewer #1 (REMARKS to AUTHOR(s)):

Comments and Suggestions for Authors

Cheng at all. using the density functional theory to investigate the structural stability, lattice parameters, electronic structure, mechanical and optical properties of six Pu oxide phases. The variations of the mechanical and optical properties in the presence of oxygen vacancies is also considered. A correlation between the oxygen / plutonium ratio and mechanical properties is found. The best optical performance is calculated for PuO. In general, the manuscript is confusing, and it is very difficult to understand which are the relevant information that the authors gain from each section. Some examples:

a   1) In section 3.2. Energy Band Structure and Electronic Density of States, the band structures and DOS/PDOS are reported but which kind of relevant information is obtained from them?

Good suggestions. In section 3.2, we calculated band structures and the density of states with spin-polarized, which are shown in the Fig. 2 and Fig. 3, respectively. From the calculated band structures and the spin state information, it is found that the band structures of ?-Pu2O3 and ?-PuO2 are only consisted of spin-up states in the range from - 2 eV to 2 eV. The other four Pu oxides are mostly composed of the spin-up electronic states in the range from -2 eV to 2 eV and also have fewer spin-down electronic states in the same range, which give us a clear point of the difference in the electronic states in these plutonium oxide compounds. We also have analyzed the contribution of each partial-wave orbital to the electronic state in detail, and found that the f orbital of Pu element has the largest contribution for all oxides, which is much higher than that of other orbitals. This makes us believe that the occurrence of f orbital near Fermi energy is an important factor for the properties of Pu oxides. Finally, we have revised this section for your better understanding. “For the metallic system, the distribution of electronic DOS will span the Fermi level and have a great impact on the properties of materials. In Figure 2, the red line indicates spin up, and the blue line indicates spin down. We can see that the spin up electron density of the six plutonium oxides crosses the Fermi level. But only the electron densities of figure 2b,2c, and 2d cross the Fermi level for spin down electrons. While for spin down electrons, the band gaps of figure 2a,2e and 2f are 4.46 eV, 4.85 eV and 4.11 eV. We have calculated the electronic projected density of states (PDOS) of plutonium oxides, as shown in Figure 3. It shows that the overall picture is similar and the electron contribution to the density of states can be discussed in three intervals from Figure 3a-f. At energies below -1 eV, for β-Pu2O3, α-Pu2O3, α-PuO2 and γ-PuO2, the DOS is mainly contributed by the p-orbitals of the oxygen atom, while the orbital contributions of γ-Pu2O3 and PuO are similar. Near the Fermi energy level, it can be clearly seen that the f-orbiting electrons of the plutonium atom play a major role and are contributed by the separate spin-up electron.  We believe that for the six plutonium oxides, the f-orbital electrons of Pu atom mainly contribute to the DOS at the Fermi level. The contribution of O atom is small, because the energy of 2p electron in the outermost layer of oxygen atom is smaller than 5f of Pu. At the same time, the 2p orbital electron of oxygen atom has a large expansibility which means the degree of electron dispersion is strong. From each figure, we can see an obvious spike in electronic density of states, which is mainly caused by the strong localization of 5f orbital electrons of Pu atom.(Page6, Line1)

b   2) In the 3.3 Section, the authors wrote “Only the β-Pu2O3 structure meets all three requirements to indicate that only β-Pu2O3 is mechanically stable.” This is the result, and then? What kind of relevant information about material properties can be derived from this sentence, and how can this sentence be related to the formation energies in Table 1? Does the fact that β-Pu2O3 is the only mechanically stable species also imply that it is the only species present under the experimental conditions? In this case, the subsequent calculations on all species are meaningless.

Thank you very much for your good question. We agree the positive suggestions of the reviewer, we have deleted this paragraph. Mechanical properties and the formation energy can confirm the stability of materials at different levels. In this paper, we use the formation energy to denote the stability of the structures. There are no stable Pu oxides at room temperature, and there is still a simulation stage for the stability of Pu oxides. In this manuscript, DFT+U method based on density functional theory was used to systematically study the structural stability, electronic structure, mechanical properties and optical properties of Pu oxides.

c   3) In section 3.3, the trends in Figure 4 are described, but what the implications are on the properties of the different trends is not reported.

Good suggestions, the optical and dielectric properties for oxides of PuO,α-Pu2O3 and α-PuO2 are discussed in previous studies[1-3], but few reports on the metastable oxides such as β-Pu2O3,γ-Pu2O3 andγ-PuO2,et.al. In Figure 4, the un-reported metastable oxide of α-Pu2O3 presents the highest dielectric properties. And light absorption spectra or reflection spectra also present different absorbed properties. We also add the discussions in the manuscript “In Figure 4, the un-reported metastable oxide of α-Pu2O3 presents the highest dielectric properties. And light absorption spectra or reflection spectra also present different absorbed properties.”. (Page9, Line7)

  1. Cary, S.K.; Galley, S.S.; Marsh, M.L.; Hobart, D.L.; Baumbach, R.E.; Cross, J.N.et alAlbrecht-Schmitt, T.E. Incipient class II mixed valency in a plutonium solid-state compound. Nat Chem 2017, 9, 856-861. https://www.ncbi.nlm.nih.gov/pubmed/28837172.
  2. Qiu, R.; Zhang, Y.; Ao, B.; Stability and optical properties of plutonium monoxide from first-principle calculation. 2017, 7. 12167. https://www.nature.com/articles/s41598-017-12428-x.
  3. Vitova, T.; Pidchenko, I.; Fellhauer, D.; Bagus, P.S.; Joly, Y.; Pruessmann, T.et alGeckeis, H. The role of the 5f valence orbitals of early actinides in chemical bonding. Nature Communications 2017, 8, 16053. https://www.nature.com/articles/ncomms16053?utm_source=xmol&utm_medium=affiliate&utm_content=meta&utm_campaign=DDCN_1_GL01_metadata.

For all these reasons I suggested to not publish the manuscript in the present form. Major revisions are eventually necessary to better highlight the aim and the implications of each calculation on the different systems.

In addition, minor revisions:

1   1) In Table 1, the formation energy unit is missing

Thank you for pointing this out. Following this suggestion, we have added units of formation energy in Table 1. (Page5)

2   2) In Figure 2, it would be better to indicate in the caption which is the atom involved in the PDOS. But in the text is indicated that Figure 2 reported total DOS, thus the authors should correct this incongruence.

I am sorry that this part was not clear in the original manuscript. Since the magnetic of plutonium oxides, it should be explained that all the electron spin states of the oxide rather than the spin states of a single element. Figure 2 does not involve the partial density of states. The data on the projected density of states should be viewed in figure 3. Finally, we have revised this section for your better understanding. “For the metallic system, the distribution of electronic DOS will span the Fermi level and have a great impact on the properties of materials. In Figure 2, the red line indicates spin up, and the blue line indicates spin down. We can see that the spin up electron density of the six plutonium oxides crosses the Fermi level. But only the electron densities of figure 2b,2c, and 2d cross the Fermi level for spin down electrons. While for spin down electrons, the band gaps of figure 2a,2e and 2f are 4.46 eV, 4.85 eV and 4.11 eV. We have calculated the electronic projected density of states (PDOS) of plutonium oxides, as shown in Figure 3. It shows that the overall picture is similar and the electron contribution to the density of states can be discussed in three intervals from Figure 3a-f. At energies below -1 eV, for β-Pu2O3, α-Pu2O3, α-PuO2 and γ-PuO2, the DOS is mainly contributed by the p-orbitals of the oxygen atom, while the orbital contributions of γ-Pu2O3 and PuO are similar. Near the Fermi energy level, it can be clearly seen that the f-orbiting electrons of the plutonium atom play a major role and are contributed by the separate spin-up electron.  We believe that for the six plutonium oxides, the f-orbital electrons of Pu atom mainly contribute to the DOS at the Fermi level. The contribution of O atom is small, because the energy of 2p electron in the outermost layer of oxygen atom is smaller than 5f of Pu. At the same time, the 2p orbital electron of oxygen atom has a large expansibility which means the degree of electron dispersion is strong. From each figure, we can see an obvious spike in electronic density of states, which is mainly caused by the strong localization of 5f orbital electrons of Pu atom.(Page6, Line1)

3  3) Authors wrote that “The total electron density of states is contributed by both spin-up and spin-down electrons, and at energies -2 eV < E < 2 eV, the total electron density of states for β-Pu2O3 and α-PuO2 is essentially contributed by spin-up electrons.” But in this range this sentence in valid for all species.

Good question, β-Pu2O3 and ?-PuO2 is only contributed by spin up states in the range from -2 eV to 2 eV, but few effect of the spin-down electronic states for β-Pu2O3 and ?-PuO2 in in the range from -2 eV to 2 eV. While other four plutonium oxide compounds are mostly composed by the spin-up electronic states in - 2 eV to 2 eV, and also a few the spin-down electronic states for other plutonium oxide compounds in the range from - 2 eV to 2 eV. We also add the discussions in the manuscript “The total electron density of states is contributed by both spin-up and spin-down electrons, and at energies -2 eV < E < 2 eV, β-Pu2O3 and ?-PuO2 is only contributed by spin up states in - 2 eV to 2 eV, but few effect of the spin-down electronic states for β-Pu2O3 and ?-PuO2 in - 2 eV to 2 eV.”. (Page5, Line15)

4  4) Authors wrote that “From Figure 2a, we can find that β-Pu2O3 has the lowest total DOS at the Fermi energy level and indicates that the structure of β-Pu2O3 is more stable compared with several other structures.” But this does not seem to be true. If the 0 in y scale on the left is the Fermi level, the lowest value of the total DOS on the x axes seems to be found for the PuO specie as clearly reported in Figure 3.

Thank you for pointing this out. Following your good advices, we have deleted these sentences. We also have revised this section for your better understanding. (Page6, Line1)

5   5) The PDOS acronymous is not “Pre-Departure Orientation Seminar (PDOS)” as reported in the caption of Figure 3. This error is particularly serious in the reviewer's opinion because it conveys the idea that it is not clear to all the authors (or at least to those who wrote this caption) what kind of calculations took place. 

We thank the reviewer for pointing out this mistake. We are ashamed of the mistake and have revised this as “projected density of states (PDOS)” immediately. Thank you again for your kindly reminding. (Page7, Line7)

6   6) In the discussion between Figures 2 and 3 authors compared “γ-Pu2O3 and γ-Pu2O3”. Probability there is a typo.

We thank the reviewer for pointing out this mistake. We have re-analyzed figure 3 again, and the original error has been deleted. Finally, we also revised the statements in this section for your better understanding. (Page6, Line1)

Reviewer 2 Report

The manuscript entitled “First-principles study of mechanical and optical behavior of plutonium oxide under typical structural phases and vacancy defects” by Jin-Xing Cheng, Fei Yang, Qing-Bo Wang, Yuan-Yuan He, Yi-Nuo Liu, Zi-Yu Hu, Wei-Wei Wen, You-Peng Wu, Cheng-Yin Zheng, Ai Yu, Xin Lu, and Yue Zhang could be of interest for the readership of the Materials journal. However, the manuscript has some significant issues including incomplete treatment of possible spin configurations and many problematic formulations that prevented me from understanding what the Authors wanted to express. The manuscript should be re-written in many places, and very careful proofreading must take place, otherwise I am unable to review it properly.

1) Major issue: It is not clear which spin states were considered. This is a vital issue as the spin choice will influence all properties that the Authors further calculate. The Authors should try various spin multiplicities and pick the most stable one for further calculations of properties.

2) The first two sentences of the abstract read: “The chemical corrosion aging of plutonium is a very important topic. It is easy to be corroded and produces oxidation products of various valence states because of its 5f electron orbit between local and non-local.” The first sentence does not provide any information. I do not understand what the Authors would like to say with the second sentence (“5f electron orbit between local and non-local”).

3) Last sentence of the abstract reads: “In terms of optical properties, PuO has the best optical properties, and the light absorption rate decreases with the increase of oxygen vacancy concentration.” What do “best optical properties” mean?

4) There is a considerable number of typos, inappropriate grammar and inactive citations in the text. The following list is incomplete, the full manuscript must be proofread: page 2, lines 3 and 4; page 3, lines 3 and 10; caption of Figure 1: “are” -> “is” (line 3), “fonts” -> “color” (line 4); first sentence in Section 3.2: “their” -> “the”; page 6: “gamma-Pu2O3 and gamma-Pu2O3 differ from…”; page 8: “And the higher the value of E, the greater of the stiffness of the material.”, “In Table 3, it shows that…”, “The reflectance spectra also well corresponds…”; page 11: “waves scatting”; page 13: “using the first nature principle”, “At an oxygen/plutonium ratio of 1.5, the elastic modulus of the γ crystal system is higher than that of the β crystal system. And the α crystal system is the smallest.”

5) It is not clear how many atoms are included in the modeled cells. I also did not find the information whether the cell parameters were optimized.

6) First few sentences of 3.1 would belong to introduction, not to the results and discussion section.

7) Page 4: The following sentence ends abruptly: “Plutonium oxide exists in a variety of compound forms, and Figure 1 shows the crystal shape structure and local”

8) Table 1: The energy unit is missing.

9) Section 3.2: “Since the plutonium oxides are magnetic, we have considered the electron spin case.” What is meant here?

10) Second paragraph of Section 3.2 is unclear and must be rewritten, it includes also incomprehensible formulations such as “The f orbital electrons of the plutonium elements have a higher probability of occurrence at the corresponding energy and are much larger than those in other orbitals of other elements.” or “This orbital electron is also a valence electron and the splitting tip indicates that the number of valence electrons is high and makes bonding easier and the material more active.” The Authors should focus on the findings they want to present – what is the insight gained through Figures 2 and 3?

11) Page 10: “There are no electronic states that are occupied between energies -4 eV and -1 eV.” I do not think the Authors calculate multiple electronic states.

12) Figure 6a: The number of decimal places in velocities seems to be exaggerated.

13) Conclusions: “But at an oxygen/plutonium ratio of 2, the α crystal system is higher than that of the γ crystal system.” What is meant here?

14) A supporting file with coordinates of all investigated cells is missing.

Author Response

----------REPLY TO THE SECOND REVIEWER-------------

Reviewer #2 (REMARKS to AUTHOR(s)):

Comments and Suggestions for Authors:

The manuscript entitled “First-principles study of mechanical and optical behavior of plutonium oxide under typical structural phases and vacancy defects” by Jin-Xing Cheng, Fei Yang, Qing-Bo Wang, Yuan-Yuan He, Yi-Nuo Liu, Zi-Yu Hu, Wei-Wei Wen, You-Peng Wu, Cheng-Yin Zheng, Ai Yu, Xin Lu, and Yue Zhang could be of interest for the readership of the Materials journal. However, the manuscript has some significant issues including incomplete treatment of possible spin configurations and many problematic formulations that prevented me from understanding what the Authors wanted to express. The manuscript should be re-written in many places, and very careful proofreading must take place, otherwise I am unable to review it properly.

1) Major issue: It is not clear which spin states were considered. This is a vital issue as the spin choice will influence all properties that the Authors further calculate. The Authors should try various spin multiplicities and pick the most stable one for further calculations of properties.

Good suggestions, we listed the energies of antiferromagnetic (AFM), ferromagnetic (FM) and nonmagnetic (NM) of the β-Pu2O3, α-Pu2O3, γ-Pu2O3, PuO, α-PuO2, γ-PuO2 in Table S1 of the supporting materials. In our manuscript, we choose the ferromagnetic (FM) ones for further calculations of properties. We also add “Spin-polarized are included in structural optimizations. Here, we consider three possibilities for the magnetic states: nonferromagnetic (NM), ferromagnetic (FM), and antiferromagnetic (AFM) for all the oxides see Table.S1. In the following FM calculations, we use the collinear 1-k structure where the atomic spin moment is along the [001] direction.” in the manuscript. (Page3, Line9)

Plutonium
oxide

AFM
    energy(eV)

FM
 energy(eV)

NM
 energy(eV)

-

-57.565

-57.803

-53.971

-

-114.427

-115.147

-106.712

-

-56.971

-57.476

-53.661

-95.555

-95.326

-88.435

-

-133.580

-134.194

-122.877

-

-66.202

-66.560

-66.373

Table S1. Energies of AFM, FM and NM for β-Pu2O3, α-Pu2O3, γ-Pu2O3, PuO, α-PuO2, γ-PuO2.

2) The first two sentences of the abstract read: “The chemical corrosion aging of plutonium is a very important topic. It is easy to be corroded and produces oxidation products of various valence states because of its 5f electron orbit between local and non-local.” The first sentence does not provide any information. I do not understand what the Authors would like to say with the second sentence (“5f electron orbit between local and non-local”).

Thank you for your question. Following your good suggestion, we have made the changes and here we want to express that plutonium has strong activity and easy oxidation. (Page1, Line3)

3) Last sentence of the abstract reads: “In terms of optical properties, PuO has the best optical properties, and the light absorption rate decreases with the increase of oxygen vacancy concentration.” What do “best optical properties” mean?

Thank you for your question. Here, it is mean that PuO has an obvious absorption peak in the visible light range, which is not found in other structures.

4) There is a considerable number of typos, inappropriate grammar and inactive citations in the text. The following list is incomplete, the full manuscript must be proofread: page 2, lines 3 and 4; page 3, lines 3 and 10; caption of Figure 1: “are” -> “is” (line 3), “fonts” -> “color” (line 4); first sentence in Section 3.2: “their” -> “the”; page 6: “gamma-Pu2O3 and gamma-Pu2O3 differ from…”; page 8: “And the higher the value of E, the greater of the stiffness of the material.”, “In Table 3, it shows that…, “The reflectance spectra also well corresponds…”; page 11: “waves scatting”; page 13: “using the first nature principle”, “At an oxygen/plutonium ratio of 1.5, the elastic modulus of the γ crystal system is higher than that of the β crystal system. And the α crystal system is the smallest.”

Thank you very much for your careful inspection. Following your suggestions, we have revised number of typos, inappropriate grammar and inactive citations in the text (yellow background in the manuscript). At the same time, we also have our manuscript checked by a native English-speaking colleague on both language and readability. We really hope that the flow and language level have been substantially improved.

5) It is not clear how many atoms are included in the modeled cells. I also did not find the information whether the cell parameters were optimized.

Thank you for your question. All our structures are optimized, and the energy convergence criteria are described in the second part. Following to the 5th and 14th opinion of reviewers, we prepared a supporting information file for structural unit cell information. The atomic number and atomic coordinates in the model can be found in the supporting materials.

6) First few sentences of 3.1 would belong to introduction, not to the results and discussion section.

Thanks for reminding, we revised them immediately. And, the statements and expressions are also revised in the section 3.1 and introduction for better understanding. (Page4, Line24, 27.., Page5, Line16)

7) Page 4: The following sentence ends abruptly: “Plutonium oxide exists in a variety of compound forms, and Figure 1 shows the crystal shape structure and local”

Thank you for your question. We revised the original text for clarity as “Plutonium oxide has multiple structures, and Figure 1 shows the crystal structure parameters and local charge density of six plutonium oxides.”. (Page4, Line27)

8) Table 1: The energy unit is missing.

Thank you for your inspection. We have added the missing phrases “energy unit” in Table 1. (Page5, Line9)

9) Section 3.2: “Since the plutonium oxides are magnetic, we have considered the electron spin case.” What is meant here?

Thank you for your question. We believe that the different spin states of electrons in the system we studied will have certain effects on the properties of materials, so we have considered the spin of electrons in our calculations. We tested the energies of antiferromagnetic (AFM), ferromagnetic (FM) and nonmagnetic (NM) of the β-Pu2O3, α-Pu2O3, γ-Pu2O3, PuO, α-PuO2, γ-PuO2 in Table S1 in the supporting materials. We also add “Spin-polarized are included in structural optimizations. Here, we consider three possibilities for the magnetic states: nonferromagnetic (NM), ferromagnetic (FM), and antiferromagnetic (AFM) for all the oxides see Table.S1. In the following FM calculations, we use the collinear 1-k structure where the atomic spin moment is along the [001] direction.” in the manuscript. (Page3, Line9)

10) Second paragraph of Section 3.2 is unclear and must be rewritten, it includes also incomprehensible formulations such as “The f orbital electrons of the plutonium elements have a higher probability of occurrence at the corresponding energy and are much larger than those in other orbitals of other elements.” or “This orbital electron is also a valence electron and the splitting tip indicates that the number of valence electrons is high and makes bonding easier and the material more active.” The Authors should focus on the findings they want to present – what is the insight gained through Figures 2 and 3?

Thank you for your question. Maybe our sentence expression is lacking, which results the lack of clarity in the meaning of the expression. We believe that the density of states represents the number of electrons allowed in the unit energy range. Because atomic orbits are mainly divided by energy, the density of states can reflect the distribution of electrons in each orbit and the interaction between atoms. The density of states can further reveal the chemical bond information, so we can draw a conclusion that the bonding of different crystal system materials is difficult. Secondly, as a visualization result of energy band, the density of states is corresponding to the energy band in many analyses. It indicates that the electron is strongly nonlocalized when the average distribution of the density of states in a certain energy range and without a local peak. On the contrary, the density of states has a large spike which indicates that the electrons are relatively local and the corresponding energy band is relatively narrow. The exactly revised expressions are as follows: “In general, the stability of the material structure was determined by the position of the value of the electronic density of states (DOS) at the Fermi energy level. For the metallic system, the distribution of electronic DOS will span the Fermi level and have a great impact on the properties of materials. In Figure 2, the red line indicates spin up, and the blue line indicates spin down. We can see that the spin up electron density of the six plutonium oxides crosses the Fermi level. But only the electron densities of figure 2b,2c, and 2d cross the Fermi level for spin down electrons. While for spin down electrons, the band gaps of figure 2a,2e and 2f are 4.46 eV, 4.85 eV and 4.11 eV. We have calculated the electronic projected density of states (PDOS) of plutonium oxides, as shown in Figure 3. It shows that the overall picture is similar and the electron contribution to the density of states can be discussed in three intervals from Figure 3a-f. At energies below -1 eV, for β-Pu2O3, α-Pu2O3, α-PuO2 and γ-PuO2, the DOS is mainly contributed by the p-orbitals of the oxygen atom, while the orbital contributions of γ-Pu2O3 and PuO are similar. Near the Fermi energy level, it can be clearly seen that the f-orbiting electrons of the plutonium atom play a major role and are contributed by the separate spin-up electron.  We believe that for the six plutonium oxides, the f-orbital electrons of Pu atom mainly contribute to the DOS at the Fermi level. The contribution of O atom is small, because the energy of 2p electron in the outermost layer of oxygen atom is smaller than 5f of Pu. At the same time, the 2p orbital electron of oxygen atom has a large expansibility which means the degree of electron dispersion is strong. From each figure, we can see an obvious spike in electronic density of states, which is mainly caused by the strong localization of 5f orbital electrons of Pu atom.”. (Page6, Line29-30, Page7, Line1-20)

11) Page 10: “There are no electronic states that are occupied between energies -4 eV and -1 eV.” I do not think the Authors calculate multiple electronic states.

Thank you for your question. From our calculation results, we did not find the case of electronic multiples. The existence of oxygen vacancy does make no electrons in the energy range occupied by O atom. Following your advice, we deleted the relevant words with doubts in the text. (Page10, Line24)

12) Figure 6a: The number of decimal places in velocities seems to be exaggerated.

Thank you for your careful observation. We have reduced some decimal places here. The reason why five decimal places are still reserved is to let the reader easier estimate and judge, and to provide more reference for subsequent calculation. (Page10, Line21)

  1. Ghosh, P.S.; Arya, A. First-principles study of phase stability, electronic and mechanical properties of plutonium sub-oxides. Phys. Chem. Chem. Phys. 2019, 21, 16818-16829. http://dx.doi.org/10.1039/C9CP01858A.
  2. Huang S.S., Ma J.J., Lai K., Zhang C.B., Yin W., Qiu R.Z., Zhang P., Wang B.T. Point defects stability, hydrogen diffusion, electronic structure, and mechanical properties of defected equiatomic γ(U,Zr) from first-principles, Materials 2022, 15, 7452-7465. https://doi.org/10.3390/ma15217452.

13) Conclusions: “But at an oxygen/plutonium ratio of 2, the α crystal system is higher than that of the γ crystal system.” What is meant here?

Thank you for your question. Here we would like to express that the elastic modulus of α crystal system is higher than that of the γ crystal system. Follow your advice, we have made this change in the text. (Page12, Line24)

14) A supporting file with coordinates of all investigated cells is missing.

Thank you for your question. According to the suggestion, we have listed the lattice coordinates in the Supporting Information files. (Page1-8)

Round 2

Reviewer 1 Report

The authors modified the manuscript with the inclusion of all suggestions. The quality of the study is definitively improved, and they dissolved my doubts on the stability of different systems. Thus, I suggest the publication in the present form.

Reviewer 2 Report

The manuscript was improved considerably during the revision process and might be accepted as is.

Minor issue, Page 3: "Spin-polarized are" -> "Spin polarization is"